# The Effects of Tamoxifen on Tolerogenic Cells in Cancer

**DOI:** 10.3390/biology11081225

**Published:** 2022-08-17

**Authors:** Ros Akmal Mohd Idris, Ali Mussa, Suhana Ahmad, Mohammad A. I. Al-Hatamleh, Rosline Hassan, Tengku Ahmad Damitri Al Astani Tengku Din, Wan Faiziah Wan Abdul Rahman, Norhafiza Mat Lazim, Jennifer C. Boer, Magdalena Plebanski, Rohimah Mohamud

**Affiliations:** 1Immunology Department, School of Medical Sciences, Universiti Sains Malaysia, Kubang Kerian 16150, Kelantan, Malaysia; 2Haematology Department, School of Medical Sciences, Universiti Sains Malaysia, Kubang Kerian 16150, Kelantan, Malaysia; 3Department of Biology, Faculty of Education, Omdurman Islamic University, Omdurman P.O. Box 382, Sudan; 4Chemical Pathology Department, School of Medical Sciences, Universiti Sains Malaysia, Kubang Kerian 16150, Kelantan, Malaysia; 5Pathology Department, School of Medical Sciences, Universiti Sains Malaysia, Kubang Kerian 16150, Kelantan, Malaysia; 6Otorhinolaryngology Department-Head & Neck Surgery, School of Medical Sciences, Universiti Sains Malaysia, Kubang Kerian 16150, Kelantan, Malaysia; 7School of Health and Biomedical Sciences, RMIT University, Bundoora, VIC 3083, Australia

**Keywords:** tamoxifen, tolerogenic cells, Tregs, Foxp3, TME, CreER system

## Abstract

**Simple Summary:**

Tamoxifen is a very well-known hormonal therapy used to treat breast cancer patients. It works by blocking the effects of estrogen in breast tissue by competing with estradiol (E_2_) in the receptor site and binding to DNA to inhibit carcinogenesis. Moreover, it is less clarified that TAM is also involved indirectly via a Foxp3 knockout model through the CreER system to target specific immune checkpoints, especially checkpoints arising in cancer therapy. The suppressive function of tolerogenic cells is very important in the TME. Hence, in our study, we observed the effects of TAM on Tregs, in which it is involved indirectly via the CreER system. In addition, we also review the effects of TAM on other cells, which are MDSCs and DCs, that act by bridging the innate and adaptive immune systems.

**Abstract:**

Tamoxifen (TAM) is the most prescribed selective estrogen receptor modulator (SERM) to treat hormone-receptor-positive breast cancer patients and has been used for more than 20 years. Its role as a hormone therapy is well established; however, the potential role in modulating tolerogenic cells needs to be better clarified. Infiltrating tumor-microenvironment-regulatory T cells (TME-Tregs) are important as they serve a suppressive function through the transcription factor Forkhead box P3 (Foxp3). Abundant studies have suggested that Foxp3 regulates the expression of several genes (CTLA-4, PD-1, LAG-3, TIM-3, TIGIT, TNFR2) involved in carcinogenesis to utilize its tumor suppressor function through knockout models. TAM is indirectly concomitant via the Cre/loxP system by allowing nuclear translocation of the fusion protein, excision of the floxed STOP cassette and heritable expression of encoding fluorescent protein in a cohort of cells that express Foxp3. Moreover, TAM administration in breast cancer treatment has shown its effects directly through MDSCs by the enrichment of its leukocyte populations, such as NK and NKT cells, while it impairs the differentiation and activation of DCs. However, the fundamental mechanisms of the reduction of this pool by TAM are unknown. Here, we review the vital effects of TAM on Tregs for a precise mechanistic understanding of cancer immunotherapies.

## 1. Introduction

Tamoxifen (TAM), a non-steroidal anti-estrogen, is a breakthrough medication for treating female breast cancer. During the course of researching contraceptives, a pharmacist unintentionally found this medication. Instead of finding new emergency contraceptives, a new potential breast cancer treatment was being discovered [1]. The first clinical trial on TAM, known as ICI46474 (beta-dimethylaminoethoxy-phenyl-l, 2-diphenylbut-l-ene), was performed in breast cancer in 1970 at the Christie Hospital in Manchester [2]. In this study, of the 46 patients that were treated, 10 showed a good response, where malignant infiltration of the skin of the chest wall regressed or healing of a malignant ulcer occurred (seven patients), with radiological resolution of pulmonary metastases (two patients) and lytic bone metastases (one patient) also noted. Clinical trials demonstrated overall that TAM has a low incidence of adverse side effects. Since then, many research and clinical trials have been conducted due to the lower incidence of side effects of TAM in breast cancer [3]. Furthermore, it looks to be a secure off-label substitute for testosterone replacement therapy (TRT) for the treatment of functional central hypogonadism in males, particularly in younger men who want to keep their fertility [4].

TAM is the oldest and most prescribed selective estrogen receptor modulator (SERM) as, overall, 75% of breast cancer patients are estrogen receptor (ER)-positive [5,6,7], while the remaining 25% are ER-negative and either exhibit overexpression of human epidermal growth factor receptor 2 (HER2) or are triple-negative breast cancers. To further understand the administration of TAM, the classification of four molecular subtypes in the 13th St. Gallen International Breast Cancer Conference is referred to (Table 1) [8].

Mechanistically, TAM has dual actions: (1) it blocks the effects of estrogen in breast tissue by competing with estradiol (E_2_) in the receptor site, and (2) it binds to DNA to inhibit carcinogenesis [9]. TAM is made to stop estrogen signaling by either inhibiting its receptor or reducing the quantity of estrogen present in the cell that can bind to other molecules [10]. If a SERM is bound to the ER, there is no room for E_2_ to attach to the cell [5]. E_2_ and estrone (E_1_), a form of weak estrogen, could bind to DNA by epoxidation, thus promoting carcinogenesis. TAM inhibits these E_2_ and E_1_ epoxides by inhibiting nuclear DNA-dependent RNA synthesis and binds to nuclear DNA and prevents the occurrence of breast cancer.

Apart from the established mechanisms, exhibiting their protective effects, as briefly outlined above, it was suggested that TAM also produces immunoprotective effects. A study showed that TAM can induce a shift from cellular (T helper 1) to humoral (T helper 2) immunity [11]. The multidrug resistance gene product, permeability glycoprotein, has lately been implicated in immunity; however, it is interesting to note that the immunomodulatory effects of TAM appear to be independent of the estrogen receptor (ER). Moreover, another study has shown that in human neutrophils, TAM stimulation increases the pro-inflammatory pathways of chemotaxis, phagocytosis and neutrophil extracellular trap (NET) production [12]. A recent article has discussed the immunomodulatory effects of TAM in the tumor microenvironment (TME) within all cells [13]. The authors suggest multifaceted immunomodulatory effects within the cells, including CD8+ T cells, CD4+ T cells, natural killer cells (NKs), dendritic cells (DCs) and neutrophils. A recent study showed that TAM exhibits anticancer effects on pituitary adenoma progression via inducing cell apoptosis and inhibiting cell migration by reprogramming tumor-associated macrophages to the M1 phenotype [14]. Besides being used as an effective hormone therapy for the treatment of hormone-receptor-positive breast cancer, TAM is also administered into animals as research tools to trigger tissue-specific gene expression in genetically modified animals based on the Cre recombinase system [15]. TAM-inducible Cre/loxP is one of the most widely used inducible systems for all gene regulations, including immune cells.

The aforementioned studies showed the immunostimulatory properties of TAM and its effects on gene regulations. However, fewer studies directly focused on the immunological tolerance regarding immune checkpoints altered by TAM on TME-infiltrating Tregs. Most of the studies are focused on alterations of specific genes in the specific tissues induced by TAM using the Cre/loxP system, which is also discussed in this review. We also focus on the effects of TAM on other tolerogenic cells. Tolerogenic cells here are referred to as immunosuppressive cells that prepare the immune system to be tolerant of different antigens, such as myeloid-derived suppressor cells (MDSCs), that are responsible for suppressing the adaptive and innate immune responses [16]. Dendritic cells (DCs) also play a critical role in modulating tolerogenic immunity by bridging innate and adaptive immunity. They kill the pathogen indirectly through the induction of long-lasting antigen-specific responses sufficiently [17]. Moreover, studies have shown that TAM also affects DCs, impairs E_2_, promotes DC differentiation and reduces the immunostimulatory capacity of DCs [18]. Hence, it is worth reviewing these cells, particularly focusing on the TME.

## 2. Immune Tolerance and Cancer

Immunological tolerance, which is the inability to react to a specific antigen, is mostly established in T- and B-lymphocytes [19]. It works as a safeguard mechanism to prevent the production of auto-reactive cells. In other words, these immunological responses are an active but carefully regulated response of lymphocytes to self-antigens [20]. To neutralize invaders, the immune system must first recognize them, which requires the immune system to be able to distinguish between self (normal cells) and non-self (invaders). The immune system can make this distinction due to the identification of molecules on the surfaces of cells. This tolerance is very important because, if breakdown occurs, it will cause autoimmunity [21].

Immunological tolerance can be subdivided into central and peripheral tolerance. It is dependent on stages during development and the function of both T and B cells. Central tolerance describes the selection processes that B cells and T cell precursors go through in the bone marrow and thymus before they are discharged as naïve T cells. Meanwhile, peripheral tolerance refers to the diverse mechanism that enforces and maintains T cell tolerance outside the thymus [22]. To further understand these two classes, Table 2 highlights the distinctions between the two tolerance mechanisms of B cell and T cell activity [20].

Lymphocytes will first be filtered by central tolerance and will be destroyed if they have high affinity to the self-reactive lymphocytes. This process is known as negative selection, where the immature lymphocytes displayed as a peptide are bound to a self-major histocompatibility complex (MHC) molecule that triggers apoptosis [23]. However, not all the immature thymocytes with high affinity will undergo apoptosis. Some of them develop into regulatory T cells (Tregs), in which a progenitor Treg population emerges that expresses a CD25hi CD4 single-positive phenotype [24]. Then, self-reactive lymphocytes, together with Tregs that can escape central tolerance, will be filtered by peripheral tolerance. The self-reactive mature lymphocytes will lead to functional inactivation, known as anergy. Anergy is an inactivation process that will be triggered by the dysfunction of lymphocytes from stimulation in the absence of costimulatory signals [25]. To achieve effective immune tolerance, all these self-reactive mature lymphocytes will not only undergo anergy. They could be eliminated through apoptosis and Treg induction, as well as deletion [22].

In cancer, immune tolerance can result from the suppression of tumor-associated antigens [26]. However, to avoid immune-mediated elimination, cancer cells may lose their antigenicity, in which the tumors stop expressing the antigens that are the targets of immune attacks. In addition, tumors employ a variety of defense mechanisms to evade immune system attack, including (1) interfering with NK cell killing by preventing them from expressing class I MHC molecules, (2) preventing T cell activation by expressing inhibitory receptors such as programmed cell death protein 1 (PD-1) and (3) activating immune checkpoints. Lastly, the tumors may avoid the destruction of the immune responses through the secretion of immunosuppressive cytokines, such as transforming growth factor β (TGF-β), or inducing Tregs that suppress the immune responses [23]. They ‘escape’ immune surveillance and later proliferate massively, forming a clinically detectable tumor [27]. Moreover, a few studies have shown that infiltrating Tregs in the TME promoted tumor growth [28] and their frequency in the tumor microenvironment has clinical significance [29].

As briefly mentioned above, Plitas (2016) demonstrated that these Tregs exhibit distinct functional and transcriptional properties compared to Tregs that reside in the blood and normal breast tissue [28]. In addition, they favor a more immunoregulatory phenotype and the tumors that they infiltrate become more aggressive. Increased-grade tumors, on the other hand, exhibited higher Treg proliferative activity to preserve their pool in the TME. By regulating the engagement to the costimulatory proteins CD80 and CD86 on dendritic cells, CTLA-4, an immunological checkpoint inhibitor and CD28 homolog, was demonstrated to play a significant role in maintaining the populations of Tregs inside the TME [30].

This showed that the involvement of Tregs, specifically with their transcriptional factors and immune checkpoints, is very complicated, especially in the TME. Hence, we further discuss Tregs’ behavior, mainly in CreER animal models, in the next section.

## 3. Utilization of TAM for Treg Regulation

Numerous factors, including cytokines and T cell receptor (TCR)/costimulatory signaling in relation to an anatomical location at steady state, affect the stability and homeostasis of T regulatory cells (Tregs). According to research, Teffs and Tregs interpret many of the same environmental cues in different ways [31]. By means of stability, Tregs are able to maintain their transcriptional factor, Foxp3, expression and resist the acquisition of pro-inflammatory effector functions during inflammation [32]. At an inflammation state, i.e., in the TME, Tregs need to integrate environmental cues and precisely tailor their activity to different immune contexts accordingly [31]. It was reported that there is a high number of Tregs and a low ratio of CD8+ T cells in the TME, which leads to Tregs’ instability [33]. Later, in 2021, it was further shown that Tregs do not only become aggressive in the TME but are actively proliferating, which therefore increases their pools [30].

On the other hand, a recent study showed that within the TME, instead of a loss of stability, ‘plastic’ Treg cells (known as Th-like Treg cells) acquire the expression of transcription factors associated with effector T cell programs by retaining Foxp3 expression [32]. This is quite confusing, where the role of Tregs in promoting the tumor growth, as mentioned above, has now become an immune tumor-suppressive function in the form of Th-like Treg cells. A recent review has stated that Tregs now serve a pivotal role as a therapeutic target within the TME by altering their cell fragility [34]. Hence, the mechanisms that mediate Tregs’ fragility and to identify essential pathways that could be targeted to induce fragile or ex-Treg cells are yet to be determined.

Therefore, it is crucial to investigate the main role of Tregs, especially within the TME, regarding whether they are fully unfavorable or might be a very effective tumor-targeted therapeutic. In the next section, we focus on the utilization of TAM for gene regulation, especially on master regulators and immune checkpoint blockers, as they are increasingly being considered for cancer treatment. As the TAM-inducible Cre/loxP system is widely used for all gene regulations, including immune cells, at the specific tissues, it gives a clearer view on the Tregs’ behavior within the TME.

### 3.1. Foxp3

Transcription factors recognized as ‘master regulators’ represent various genetic programs directing precursor cells to a specific cell lineage. In regard to Tregs, Forkhead box P3 (Foxp3) is exclusively expressed on Tregs and contributes to their suppressive function, as strongly evidenced in a severe, rare X-linked fatal autoimmune disease known as immune dysregulation polyendocrinopathy enteropathy, X-linked (IPEX) syndrome, where mutations of this transcription factor occur [35].

Numerous animal models support the significant role of Foxp3 in Tregs as both phenotypic and functional markers. The early support of the role of Foxp3 in dominant tolerance came from the observations of elevated levels of Foxp3 in peripheral and thymic CD25+ Treg cells in mice. On the other hand, Foxp3 expression was diminished in CD25+ conventional T cells activated under physiologically related circumstances [36,37,38,39]. In addition, the expression of Foxp3 in CD25+CD4+ Tregs in wild-type mice and the reduced numbers of CD25+CD4+ Tregs in scurfy and Foxp3-knockout mice hinted at a role of Foxp3 in the development of Tregs [37,38,39]. In humans, Foxp3 is also highly expressed on CD25+CD4+ T cells with a suppressive function, although Foxp3 is slightly produced in activated conventional T cell populations in humans [40,41]. Only a few human T cells upregulate Foxp3 after activation; this upregulation is temporary and involves notably lower amounts of Foxp3 than those detected in human Treg cells [42].

In cancer, Foxp3 is shown to mediate tumor immune escape [43,44]. Foxp3 has been described to regulate the expression of several genes involved in carcinogenesis to utilize its tumor suppressor function [45]. In addition, Foxp3 contributes to the regulation of breast cancer metastasis by downregulating the expression of some metastasis-associated molecules, such as CXCR4 and CD44 [46,47].

Regarding TAM, it was first used as a therapy agent approved for breast cancer treatment in the year 1977. It binds to ER, hence subsequently stopping the cell cycle in the G0 and G1 phases, thus stopping cell division [48]. In the investigation of specific Foxp3 expression on Tregs via the Cre/loxP system, TAM was used to regulate the *Foxp3* gene in the Foxp3 knockout model, known as Foxp3^YFP-Cre^. Prior to being given TAM, a FoxP3 knockout model was first created by introducing a complementary DNA (cDNA) encoding an enhanced green fluorescent protein (eGFP) with Cre recombinase and a mutated human estrogen receptor ligand binding domain (eGFP-Cre-ERT2) fusion protein into the 3′ untranslated region (UTR) of the *Foxp3* gene. These mice were then crossed with the ROSA26 locus, which contained (YFP) [43]. After the breeding, the model will transform into the Foxp3^eGFP-Cre-ERT2^ x R26Y mice model. The Foxp3^eGFP-Cre-ERT2^ x R26Y model is not able to express YFP since the GFP-CreERT2 fusion protein is isolated in the cytosol due to the locus containing a loxP site-flanked STOP cassette [41]. This is when TAM plays its crucial roles via the Cre/loxP system by allowing the nuclear translocation of the fusion protein, the excision of the floxed STOP cassette and the constitutive and heritable expression of YFP in a cohort of cells that expressed Foxp3 at the time of TAM administration. It is important to note that the utilization of TAM is crucial in the Foxp3 knockout model as, without the involvement of TAM, the stability of Foxp3 cannot be evaluated.

Additionally, TAM also played a significant role in developing Foxp3 conditional knockout mice (FoxP3floxR26CreERT2) [49]. Breeding mice with a floxed allele of the *Foxp3* gene (Foxp3flox) and Cre recombinase-estrogen receptor 2 fusion protein (CreERT2) from the ROSA26 locus led to the development of the model (R26CreERT). The conditional Foxp3 knockout mice (Foxp3floxR26CreERT2) produced were then induced with TAM, in which the mechanism of action also involves the Cre/loxP system. Different to the study mentioned above [43], instead of using TAM to express FoxP3, this study induced TAM to delete the loxP-flanked allele of the *Foxp3* gene to access the humoral immunity [49], as Foxp3+ Tregs are part of adaptive immunity. As a result, the knockout model Foxp3 with the *Foxp3* gene deleted had higher serum levels of immunoglobulins, including autoantibodies such as anti-dsDNA antibodies. Moreover, this model also induced T and B cell activation, particularly T follicular helper (Tfh) cells and germinal center (GC) B cells [49]. Hence, this study suggests that Tregs are very important to control excessive or pathogenic antibody production by inhibiting unnecessary Tfh differentiation and the subsequent GC responses, which are achievable by the involvement of TAM. As Foxp3 is the master regulator for Tregs, its expression is indeed very stable under steady-state conditions [50], which is in line with the aforementioned study using the Foxp3 conditional knockout mice (FoxP3floxR26CreERT2) system [49]. Tai and colleagues reported that in Foxp3^GFP-Cre-ERT2^ mice, after being treated with TAM for 3 days as adults and examined 2 weeks or 5 months later, <5% of YFP+ cells were found to be negative for Foxp3 expression.

Another critical study on the Foxp3 knockout model focused on generating animals specifically lacking Rieske iron–sulfur protein (RISP) in Treg cells (RISP KO). This model was created to test whether the mitochondrial respiratory chain complex III is necessary for Treg cells’ survival, proliferation or function [51]. The model was developed by breeding Foxp3YFP-Cre mice with the loxP-flanked *Uqcsrf1* gene, which encodes the critical subunit of mitochondrial complex III, known as Rieske iron–sulfur protein (RISP), to produce animals that are selectively deficient in RISP in Treg cells (RISP KO). Then, mice harboring the Uqcrq floxed were again generated between Foxp3^GFP-CreERT2^ and ROSA26SorCAG-tdTomato alleles (QPC iKO) to study the loss of complex III after development impairs Treg cell function. In these animals, GFP marks cells actively expressing Foxp3, while tdTomato red fluorescent protein (RFP) identifies cells that have undergone the Cre-recombinase-mediated loss of Uqcrq. TAM was administered to delete the loxP-flanked allele of the Foxp3 gene, and the result showed that there was systemic inflammation and significant elevations in activated CD4+ and CD8+ T cells [51]. This work therefore suggested that Treg-cell-specific deletion of mitochondrial respiratory chain complex III results in the early onset of a deadly inflammatory illness, without affecting the number of Treg cells. However, Treg cells require mitochondrial complex III to maintain their immune regulatory gene expression and suppressive function.

In cancer, the TAM-inducible Foxp3 knockout model is also being used to study Tregs. For example, in a study of Interferon regulatory factor 4 (IRF4) with effector Treg differentiation and immune suppression in cancer, Foxp3^EGFP-Cre-ERT2^ mice were crossed with Irf4-floxed mice to allow for the specific deletion of *Irf4* in Foxp3+ cells following TAM treatment [49]. IRF4 is a subset of intratumoral CD4+ effector Tregs with superior suppressive activity. The study showed that the induced deletion of *Irf4* in Foxp3+ cells in MC38 tumor-bearing mice resulted in a significant delay in tumor growth. Hence, the study concluded that IRF4+ Tregs suppress antitumor immunity. Furthermore, another recent study also used the TAM-inducible Foxp3 knockout model to study the expression of T cell immunoglobulin and mucin domain-containing protein 3 (Tim-3) in the TME [49]. They use the same model from the previous study, which was the flox-stop-flox *Tim3* (FSF-Tim3) mouse model, to drive Tim-3 expression in a Cre-dependent manner [52]. The model was then crossed with Foxp3^EGFP-Cre-ERT2^ and TAM was administered to delete *Tim3* in FoxP3+ cells. Thus, they proposed that Tim-3’s inducible deletion, unique to Tregs, improves antitumor immunity. Interleukin-10 (IL-10) expression and a change to a more glycolytic metabolic profile are both substantially linked with Tim-3’s enhancement of Treg cell function [53].

Overall, the aforementioned studies showed that TAM plays a significant role in investigating Tregs, specifically through the *Foxp3* gene knockout animal model, especially in mice, by means of targeting genes that are disrupted or inactivated. These animal studies are crucial as animals can mimic aspects of a disease found in humans. In general, the role of TAM used in the Foxp3 knockout model is dependent on the estrogen receptor (ER) system in gene targeting. In principle, one mouse must have a tissue-specific driven Cre gene and another mouse must have loxP flanked (floxed) alleles of the interest gene. Expression of Cre recombinase excises floxed loci and inactivates the gene of interest (GOI) [54]. Hence, in the use of TAM, ER must be attached to the Cre gene to allow the excision of a chromosomally integrated GOI [54].

For example, gene targeting by CreER was used in the study mentioned above by Tai (2019), in which the tissue-specific target driven by the Cre gene was R26CreERT while the floxed allele was Foxp3flox. As result, the expression of Cre recombinase excised FoxP3flox with the help of TAM, coinciding with a study of the humoral immunity regarding the dysregulation of FoxP3. In addition, Rudensky (2011) also applied this type of gene targeting to further understand Tregs’ stability. Thus, the study chose to express the FoxP3 gene instead of inactivating the GOI. In the study, the tissue-specific target driven by the Cre gene was *Foxp3^eGFP-Cre-ERT2^*, while the floxed allele was the ROSA26 locus containing a loxP site-flanked STOP cassette. As result, the ROSA26 locus containing a loxP site-flanked STOP cassette was excised and FoxP3 was expressed with the help of TAM. In short, TAM plays a role in the Foxp3 knockout model with the genes targeted by CreER. However, the advanced effects of TAM on different Foxp3 knockout models may vary in the prospective study of interest. The most important aspect in the Foxp3 knockout model is the understanding of how the TAM-inducible system of CreER works.

In cancer, the area of interest is not only the expression or the inhibition of Foxp3. In fact, the knockout models are more specific to various immune checkpoints. For example, the studies mentioned above [53,55] focused on specific immune cells such as IRF4 and Tim-3. This is because the specific blockade of immune checkpoints can kill cancer cells better [7]. Currently, the functions of immune checkpoints to suppress the TME are designed to target the receptor–ligand interaction [56]. There are many targeted receptor–ligands, as they have shown the promising potential of immune checkpoint inhibitors (ICIs) in suppressing the TME (Figure 1). A recent review stated that the main immune checkpoints on tumor-infiltrating and peripheral Tregs are CTLA-4, PD-1/PD-L1, LAG-3, TIM-3 and TIGIT [56]. However, among all the immune checkpoints stated, not all of them involve inducible TAM. Only very few studies have used TAM in the disruption of immune checkpoints such as CTLA-4 and PD-1, which will be discussed further in the next section.

### 3.2. CTLA-4

CTLA-4 is a protein found on T cells that aids in the regulation of the body’s immune responses. The disruption of this protein can be associated with many autoimmune diseases due to impaired inhibitory checkpoint and Treg function, leading to loss of B and T cells’ stability [23]. The binding of CTLA-4 to another protein expressed on antigen-presenting cells (APCs), such as the B7 family (CD80, CD86, CD274, CD275, CD276, CD273, CD277), keeps the T cells in the inactive state, rendering them unable to kill other cells, including cancer cells [57]. CD28 is a homologous receptor expressed by both CD4+ and CD8+ T cells that mediates opposing functions in T cell activation [58]. Both receptors share a pair of B7 ligands expressed on APCs. In short, the binding of CTLA-4 with the B7 family leads to an inactive state of T cells, while the binding of the B7 family with CD28 will activate T cells [59]. Of note, CTLA-4 is abundantly expressed on Tregs. Jain (2010) reported that CTLA-4 plays a dual role in preserving T tolerance in Tregs. Functions include preventing self-reactive pathogenic T cells from accumulating harmfully in key organs and preventing incorrect naive T cell activation in conventional T cells [60]. Additionally, by reducing the amount of CD80/86 ligands available for the positive costimulation of CD28 in effector T cells, CTLA-4 in Tregs has been shown to indirectly and cell-extrinsically decrease T cell activation [61].

A recent study by Marangoni (2021) showed that, through a CTLA-4 and CD28-dependent feedback loop, Treg cells self-regulate to adapt their population size to the level of local costimulation [30]. They emphasize that Tregs act as a biological rheostat on conventional dendritic cells (cDCs) by controlling the engagement with costimulatory proteins CD80 and CD86 via CTLA-4 or CD28. Engagement of CD28 results in an intratumoral pool of hyper-proliferative Tregs that are deprived of their use of CTLA-4, but maintain their immunosuppressive function [30].

Through the use of inducible TAM, the *ctla4* gene was altered for this investigation. In order to create a paradigm where a portion of the Treg cell population may be acutely made CTLA-4-deficient by TAM therapy, bone marrow chimaeras (BMCs), Foxp3^creERT2^ x Ctla4^f/f^ and wild-type (WT) donor bone marrow implanted into CD45.1 WT hosts (Foxp3^creERT2^ x Ctla4^f/f^ BMCs) were crossed. The results showed that the deficiency of CTLA-4 does not affect the expression of Foxp3. Hence, the authors developed another model by altering *CD28* to determine the correlation between these two ligands by crossimplanting *Foxp3^creERT2^* x *Cd28^f/f^* mice, which then were treated with TAM to delete *CD28* in tumor-infiltrating Treg cells. As result, there was also no effect on the expression level of Foxp3. However, this model manages to demonstrate that CD28 is required to sustain Treg cell proliferation in the TME [30].

Based on the recent study mentioned above, we can agree that the functions of Tregs via CTLA-4 in maintaining stability within the TME are volatile. They may change accordingly, instead of undergoing a loss of stability, supporting the plasticity in Tregs. Thus, we highlight that the main role of Tregs via CTLA-4 is to expand based on their environment, even though their migration has no significant effects. To support this statement, the latest study by Zappasodi (2021) also emphasizes Tregs’ plasticity via CTLA-4, mainly in glycolysis metabolism, using the TAM-inducible model. They found that CTLA-4 blockade reduced tumor competition for glucose, which may make treatment easier, encouraging its use in conjunction with tumor glycolysis inhibitors. Hence, the study acknowledges that the effect of CTLA-4 blockade on the destabilization of Treg cells is dependent on Treg cell glycolysis and CD28 signaling by mimicking the highly and poorly glycolytic states within the TME [62].

In summary, CTLA-4 is an endocytic molecule that binds to two distinct ligands, CD80 and CD86, inactivating them. CTLA-4, on the other hand, can function as an immune-regulatory mechanism by directly reducing APC’s ability to stimulate via CD28 [58]. However, this deduction is only made from a few studies, since there is very little research investigating CTLA-4 via the CreER mice model within the TME. This concept of the regulation of CTLA-4 via Tregs helps to explain why a stimulatory and inhibitory receptor share the same ligands and embraces the endocytic nature of CTLA-4. Hence, we highlight that the CTLA-4 gene via Tregs functions as a rheostat that can tune T cell activation upwards or downwards.

### 3.3. PD-1

Aside from CTLA-4, PD-1 is another well-defined immune checkpoint, which is a protein found on T cells and helps to keep the body’s immune responses in check. This immune checkpoint was originally called programmed cell death protein 1 (PD-1), since it is thought to be involved in programmed cell death, but now it is known not to have a role in cell apoptosis [63]. However, although PD-1 has no involvement with programmed cell death, a study showed that the induction of PD-1 on activated T cells causes T cells to ignore cancer cells as one of the ‘self’ components, thus preventing T cells from becoming cancer cells [64]. Hence, PD-1 is not involved in programme cell death directly but plays an important role in self–non-self discrimination. PD-1 recognizes two ligands to inhibit T cells. The two ligands are (1) PD-L1, which is expressed on APCs and many other tissue cells, and (2) PD-L2, which is expressed mainly on bone marrow-derived APCs [63].

Of note, PD-1 is abundantly expressed on Tregs. PD-1 has been shown to inhibit Treg function by increasing PI3K/Akt/mTOR signaling activity, implying that PD-1 is an unbiased inhibitor of T cell activities, suppressing both the effector and regulatory functions depending on cell type [65]. The best-studied interaction is between PD-1 and its ligand, PD-L1 and the immune checkpoint inhibitor PD-1/PD-L1 that inhibits T cell activation and promotes T cell cycle arrest, anergy and apoptosis [66]. PD-1 is majorly expressed on the T cells of the immune system, whereas PD-L1 is expressed on cancer cells and APCs [67]. Cai (2019) reported that the relationship between the PD-1/PD-L1 pathway and Tregs has not been fully explained. However, the PD-1/PD-L1 axis exerts crucial roles in regulating Tregs’ development and function [68]. The functions of PD-1/PD-L1 are as follows: (1) PD-L1 increases Foxp3 expression and enhances the immunosuppressive ability of Tregs, and (2) PD-L1 could convert naive CD4+ T cells to Tregs through the downregulation of protein kinase B (Akt), mammalian target of rapamycin (mTOR) and protein sequences of ERK2 (ERK2) and the simultaneous upregulation of phosphatase and tensin homolog (PTEN) [68].

With the involvement of the TAM-inducible Cre model, a recent study found that cell-intrinsic PD-1 restraint of Tregs is a significant mechanism by which PD-1 inhibitory signals regulate T cell tolerance and autoimmunity [69]. In the study, the authors crossed Pdcd1^f*l/fl*^ mice with Foxp3^ERT2^.Cre mice (referred to as iFoxp3Cre Pdcd1*^fl/fl^* mice) to determine how PD-1 controls Tregs’ functions. The *PD-1* gene was then selectively deleted in Tregs following TAM administration. As a result, they described that the suppressive capacity of PD-1-deficient Tregs was caused by the reduced signaling through the PI3K–AKT pathway as a mechanism. However, this study did not focus on the TME, but on ameliorated experimental autoimmune encephalomyelitis (EAE) and protection from diabetes in nonobese diabetic (NOD) mice lacking PD-1 selectively in Treg cells.

In conclusion, PD-1 is very well investigated in terms of its ligand, PD-L1, due to its crucial roles in the activation of T cells. The inhibition of this ligand resulted in the immune suppression of Tregs via the reduced signaling of the PI3K–AKT pathway as a mechanism. However, there is a lack of study of Tregs via PD-1 within cancer. Of note, there is no specific study using the inducible TAM Cre animal model in correlation with Tregs via PD-1. Thus, there is a need to conduct such a study to obtain a sufficient explanation of the roles of Tregs via PD-1, specifically in the TME, as this study might lead to establishing potential immune therapeutics.

### 3.4. Other Immune Checkpoints

Other than the two main immune checkpoints mentioned above (CTLA-4 and PD-1), there are still other immune checkpoints that are considered as crucial checkpoints in investigating tumor-infiltrating and peripheral Tregs. These other immune checkpoints are considered as the next wave of co-inhibitory receptor targets in cancer therapy [70]. The immune checkpoints are LAG-3, TIM-3, TIGIT and TNFR2 [71,72]. These immune checkpoints play significant roles in examining the role of Tregs within the TME. However, there are very limited/no studies focusing on all these immune checkpoints, especially using Cre animal models within the TME.

LAG-3

Lymphocyte activation gene 3 (LAG-3) is one of the crucial immune checkpoints. LAG-3 can be found on the cell surfaces of Teffs and Tregs to control the T cell response, activation and growth [73]. In immune tolerance, the expression of LAG-3 is very important as the increase in this receptor protein will turn off the immune response, so that the T cell does not go on to attack healthy cells. The binding of LAG-3 to a type of antigen-presenting complex called MHC II keeps the T cells in the inactive state, inhibiting them from being able to kill other cells, including cancer cells. However, a recent study found that LAG-3 may bind to another four ligands, which are FGL-1, α-synuclein fibrils (α-syn) and the lectins galectin-3 (Gal-3) and lymph node sinusoidal endothelial cell C-type lectin (LSECtin) [74]. As well as MHC II, the binding of LAG-3 with FGL-1, Gal-3 and LSECtin has been shown to induce the inhibition of T cell activation, except for α-syn, in which the impact has not been well studied [74].

LAG-3 is abundantly expressed on Tregs. LAG-3 acts as a signal to Treg populations and contributes to their suppressor activity [75]. Moreover, LAG-3 intrinsically limits Tregs’ proliferation and function at inflammatory sites [76]. Cancer patients’ tumor locations are populated by LAG-3-expressing Tregs and fatigued cytotoxic T lymphocytes. Inhibiting LAG-3 may allow T cells to regain their capacity for cytotoxicity and potentially slow the growth of tumors, according to preclinical research [77]. In the TAM-inducible Cre model, a study demonstrated that ADAM-mediated cell surface shedding of LAG3 is important for effective antitumor immune responses [78]. This study, using a conditional knock-in mouse model, suggested that LAG-3 acts as a mechanism of primary resistance with advanced cancer receiving checkpoint blockade therapy. Lag3NC.L/L mice were then crossed with several Cre recombinase mouse lines to facilitate cell-type-restricted deletion to all T cells (CD4Cre), CD8+ T cells (E8ICre.GFP), CD4+ T cells (TAM-inducible ThPOKCreERT2) or regulatory T cells (Tregs; TAM-inducible Foxp3CreERT2.GFP). Of note, there is no other study focusing on LAG-3 using the TAM-inducible Cre model within the TME.

TIM3

The TIM family member T cell immunoglobulin and mucin domain-containing protein 3 (TIM3) was first discovered as a receptor expressed on interferon-producing CD4+ and CD8+ T cells [79]. It was reported that the murine genome contained eight predicted Tim genes, four of which (TIM1-TIM4) encode functional proteins, whereas the human genome only contains three Tim genes (TIM1, TIM3 and TIM4). As discussed below, specific ligands have been described for the IgV domains of TIM-1 and TIM-3, although no ligands have yet been described for the respective mucin-like domains [80]. TIM3 is also highly expressed in Tregs. The value of TIM3-targeting treatment techniques against cancer is increased because TIM3 designates tissue-resident Tregs that are highly suppressive and play a significant role in determining the antitumor immune response in situ [81]. However, a significant portion of TIL Tregs express a negative regulator of Th1 immunity; however, it is yet unknown how functionally active TIM-3+ Tregs are [82]. Research by Gautron (2014) reported that TIM-3 expression on Treg cells identifies a population highly effective in inhibiting pathogenic Th1 and Th17 cell responses [83].

In the TAM-inducible Cre model, the TAM-inducible Foxp3 knockout model has been used to study the expression of T cell immunoglobulin and TIM3 in the TME [53]. The authors used a flox-stop-flox TIIM3 (FSF-TIM3) mouse model to drive TIM-3 expression in a Cre-dependent manner. The model was then crossed with Foxp3^EGFP-Cre-ERT2^ and TAM was administered to delete TIM3 in FoxP3+ cells. Thus, they suggested that the Treg-specific inducible deletion of TIM-3 enhances antitumor immunity. The enhancement of Treg cell function by TIM-3 is strongly correlated with the increased expression of interleukin-10 (IL-10) and a shift to a more glycolytic metabolic phenotype [53].

TIGIT

T cell immunoreceptor with Ig and ITIM domains (TIGIT) is an important inhibitory molecule within the PVR/nectin family, and is associated with human cancers and T cell exhaustion phenotypes. The inhibition of TIGIT can enhance antitumor T cell responses through its role as a ligand, receptor and competitor for the costimulatory receptor CD226 [83]. Foxp3+ T cells express the co-inhibitory molecule TIGIT as a distinct Treg subset that specifically suppresses pro-inflammatory Th1 and Th17 cells, but not Th2 cell responses [83]. The connection of TIGIT on Tregs induced the expression of the effector molecule fibrinogen-like protein 2 (Fgl2), which promoted the Treg-cell-mediated suppression of Teff cell proliferation. In addition, Fgl2 was necessary to prevent the suppression of Th2 cell cytokine production in a model of allergic airway inflammation.

In the TAM-inducible Cre model, a conditional knockout model has been used to study the functional synergy between PD-1 and TIGIT in antitumor immunity and autoimmunity [84]. MC38 was used to develop tumors in ERT Cre mice and TAM was administered on T cells to assess PD-1 and TIGIT. The research reported that inhibiting PD-1 and TIGIT does not cause more severe disease as compared to inhibiting PD-1 alone. Other than this mentioned study, no other vital research has been done in investigating TIGIT using the Cre model within the TME. Hence, it is crucial to undertake this research thoughtfully to address this research gap.

TNFR2

Tumor Necrosis Factor 2 (TNFR2) is derived from Tumor Necrosis Factor (TNF). TNF can be present in two different forms, which are known as the membrane-bound form of TNF (mTNF) and soluble form of TNF (sTNF) [72]. Moreover, it was also reported that both forms vary in their biological actions, e.g., mTNF has been shown to mediate inflammatory responses in astrocytes, but not in neurons; meanwhile, sTNF has similar proinflammatory effects in both cell types [85]. Both sTNF and mTNF are regulated by binding with their two receptors localized at the cellular surface, TNFR1 and TNFR2. The interaction of TNF-TNFR2 on MDSCs is reported as a vital accelerator of cell development and suppressive effects [86]. Since it has been reported that TNF is implicated in tumor development and immune invasion upon interaction with its receptor 2 (TNFR2), TNFR2 has shown preferential expression on cancer cells and immunosuppressive cells [72,85].

Additionally, TNFR2 has become one of the vital immune checkpoints in cancer treatment. Within the TME, TNFR2 has been reported to directly promote the occurrence and growth of some tumor cells, activate immunosuppressive cells and support immune escape [87]. Hence, this makes TNFR2 one of the crucial targeted markers for cancer treatment. Moreover, TNFR2 is abundantly expressed on Tregs and is essential for Treg expansion and function maintenance through the classical NF-κB pathway. High expression of TNFR2 is a feature of tumor-associated Tregs, which effectively suppress the antitumor immune responses in a variety of cancer types [88]. In regard to the Cre model, no research has been done to investigate the roles of TNFR2 associated with Tregs with the administration of TAM within the TME. Hence, this indicates that there is still a research gap in identifying the involvement of TAM in exploring the roles of TNFR2 within the TME.

Overall, the involvement of all immune checkpoints (CTLA-4, PD-1, LAG-3, TIM3, TIGIT, TNFR2) associated with Tregs has shown significant inhibition in T cells and their suppressive activity. While their expression on regulatory T (Treg) cells ensures the correct functioning of Treg cells to control effector T cells, their expression on effector T cells ensures proper contraction of effector T cell responses. The TME synergizes various immune cascades to antagonize the immune-mediated elimination of their existence. Therefore, immunological checkpoints, however, are only one part of a complex network of regulatory processes [89]. As briefly mentioned above, other tolerogenic cells are also involved in antagonizing the TME’s immune-mediated elimination. Below, we further dissect how TAM exhibits effects on MDSCs and thus affects the TME.

## 4. Myeloid-Derived Suppressor Cells (MDSCs)

Myeloid-derived suppressor cells (MDSCs) are a type of heterogeneous immature myeloid cells that are responsible for suppressing the adaptive and innate immune responses [16]. Until the early 1970s, there were no studies assessing the suppressive immune effects of myeloid cells [90]. After more than 10 years, Lopez and her team reported for the first time a few myeloid cell populations that were implicated somehow in the suppression of the immune response in cancer [91]. In the late 1990s, Gr-1+CD11b+ cells were reported as immune-suppressive myeloid cells in tumor mouse models, and later, extensive studies were conducted to study these cells and their mechanisms of action [92,93]. Finally, in 2007, these cells were given the name MDSCs, and since then, there has been growing interest in learning more about the suppressive function of MDSCs [94] (Figure 2).

MDSC-mediated immunosuppression occurs by various pathways, including (1) direct interactions between MDSCs and T cells, (2) production of inhibitory cytokines, prostaglandin E_2_ (PGE2), adenosine, myeloid-related proteins (MRP8/14) and free radicals, both reactive oxygen species (ROS) and reactive nitrogen species (NOS), (3) degradation of essential amino acids (arginine and cysteine) and (4) the indirect inhibitory effect of ADAM metallopeptidase domain 17 (ADAM17) [95,96,97]. The overall mechanisms of these four pathways are summarized in Figure 2.

The inhibitory cytokines interleukin-10 (IL-10) and transforming growth factor beta (TGF-) are, however, produced by MDSCs. These cytokines induce the expansion of regulatory T cells (Tregs), as major immunosuppressive cells, and the recruitment and polarization of M2 macrophages, which also promotes IL-10 and inhibits IL-12 production [98]. Increases in IL-10 production showed an inhibitory effect on the activation and differentiation of Teffs and inhibited their cytokine production [99]. IL-10 also has an inhibitory effect on DCs, which results in a reduction in IL-12 secretion and maturation, inhibition of costimulatory molecule expression (CD40, CD80 and CD86) and promotion of immunosuppressive molecules [100]. Moreover, IL-10 inhibits the proliferation and cytotoxicity of natural killer (NK) cells and downregulates the transmembrane proteins NKG2D and NKp30 [101,102].

Moreover, it has been reported that the production of PGE2 is positively correlated with the inhibition of NK cells, which results in a decrease in cytolytic activity on NK cells and the secretion of interferon gamma (IFN-ɣ) [103]. PGE2 also contributes to the inhibition of DCs and Teffs, which results in reducing IL-2 synthesis, IL-2R expression and Th1 differentiation [104]. However, PGE2 induces the expansion, differentiation, accumulation and suppressive activity of Tregs and MDSCs [104]. The extracellular production of adenosine is also involved in MDSC-mediated immunosuppressive effects [96]. Adenosine inhibits the activity of both Teffs and NK cells, the maturation of DCs and the secretion of IL-12, while expanding Tregs and activating M2 macrophages [96]. Another vital product in MDSC-mediated immunosuppression is the myeloid-related protein (MRP, also known as S100) family, including MPR8 and MPR14 (also known as S100A8 and S100A8A9, respectively) [105]. The MPR8/14 calprotectin promotes the accumulation of MDSCs, attracts MDSCs to the tumor microenvironment and reduces the precursor differentiation of myeloid cells [106].

Oxidative stress is a result of an imbalance between free radicals and antioxidants; it involves the development of several abnormal molecular and cellular-related conditions [107,108]. It has been reported that the generation of reactive oxygen species (ROS) and reactive nitrogen species (RNS) from MDSCs inhibits the proliferation and migration of CD4+ and CD8+ T cells and reduces IL-2 expression [97,109]. Furthermore, studies showed that L-arginine and L-cysteine synergistically promote the survival, development and antitumor activity of T cells [110,111]. Therefore, the depletion of L-arginine and L-cysteine by MDSCs results in the reduced proliferation and activity of T cells through the inhibition of the T cell receptor (TCR)-CD3 zeta chain, which plays essential roles in the expression, structure and function of T cells [112]. This depletion also reduces the proliferation and activity of NK cells, while it induces the development and proliferation of Tregs [110]. Furthermore, by downregulating CD62L (L-selectin), which prevents the trafficking and activation of naive T cells, ADAM17 indirectly contributes to MDSC-mediated immunosuppression by limiting the ability of naive (CD4+ and CD8+) T cells to travel from their homes to sites where they are activated (lymph nodes) [113].

### Effects of Tamoxifen on MDSCs

In the study of the effects of TAM in MDSCs, one study reported that TAM triggered the creation of neutrophil extracellular traps (NETs), chemotaxis and other pro-inflammatory pathways in human neutrophils [12]. All these pro-inflammatory pathways are the result of the modulation of intracellular ceramides. For several processes, including apoptosis, cell development, differentiation, senescence, diabetes, insulin resistance, inflammation, neurodegenerative diseases or atherosclerosis, intracellular ceramides serve as secondary messengers [114]. In the case of TAM-induced ceramide, it has contributed to innate immunity. This has been achieved through the accumulation and mediation of certain ER-independent effects of the drug against cancer tissues, hence becoming potent inducers of NETosis, a novel cell death pathway [12].

In addition, circulating leukocyte populations and systemic treatment effects in patients with metastatic breast cancer indicate that TAM tends to enrich NK and natural killer T (NKT) cells in the circulation, whereas both chemotherapy and endocrine therapy reduce the levels of circulating monocytic MDSCs (Mo-MDSCs) [115]. This shows that the systemic immunosuppressive profile seen in patients tends to improve over the course of systemic therapy and offers hope for future immunotherapy treatments used in combination with conventional antitumor medicines. This study displayed continuous increased levels of Mo-MDSCs, where their level was usually reduced. This is consistent with earlier research that showed that the in vitro treatment of primary human monocytes with docetaxel decreased the number of Mo-MDSC-like cells while increasing the production of pro-inflammatory M1 macrophages [116].

To conclude, MDSCs are vital in the suppressive function not only within the TME, but also in a wide spectrum of antimicrobial activity [117]. In summary, TAM mainly acts as a host-mediated protective system able to kill microbial cells within innate immunity. Remarkably, despite the therapeutic and pharmacological consequences of this mechanism of action, the activity of TAM has not been well studied in innate immune cells such as macrophages.

## 5. Dendritic Cells (DCs)

Behjati and Frank (2009) have reviewed studies that demonstrate the immunomodulatory effects of TAM and suggested that this anti-estrogen can shift cellular Th1 responses to humoral Th2 immunity [11]. Cytotoxic T lymphocytes (CTL) appear to protect against tumor development in acute tumor-directed immune responses, whereas humoral immunity activation and infiltration by Th2 cells further increase tumor development and disease progression in chronic immunological responses [118].

Dendritic cells (DCs) are among the innate immune cells that serve as the first line of defense against pathogens and foreign antigens, including tumor antigens. Upon perturbation of tissue homeostasis, DCs take up the tumor antigens and migrate to the lymphoid organs, where they present their loads to induce adaptive T and B cells [117]. The outcomes of this interaction depend on DCs’ maturation status, where the presentation of antigens by inactivated DCs leads to tolerance, while mature DCs induce antigen-specific immunity. Adequate presentation of tumor antigens by costimulatory molecules of DCs such as major histocompatibility complex (MHC) classes I (MHC-I) and MHC-II generates CTL responses against tumor cells, as shown by the effective antitumor CTL by antigen-pulsed DCs [119,120]. In invasive breast cancer patients, their peripheral blood DCs are markedly reduced, with a more mature phenotype compared to healthy controls [121].

Infiltrating DCs in breast cancer are shown to promote CD4+ T cells in secreting IL-13, which promotes early tumor development [122]. Plasmacytoid DCs (pDCs), a subset of DCs well known for their antiviral immunity through the major production of IFN, are densely accumulated in aggressive breast tumors and produce very low amounts of IFN in vitro [123]. The deficiency of IFN production in these tumor-associated DCs is involved in the sustenance of Foxp3+ Tregs’ expansion, in which its increment is in line with the disease stage and provides immunosuppression for tumor cells.

### Effects of Tamoxifen on Dendritic Cells

Around 75% of breast cancers express ER; hence, treatment with selective ER inhibitors such as TAM serves as the first line therapy in breast cancer. TAM acts as both an agonist and antagonist of ER signaling dependent on the tissue and cell type. Since DCs also express ER, TAM’s suppression of ER during cancer therapy reduces DC development and activation [17,102,123]. TAM inhibits the differentiation into DCs through the dysregulation of signature markers such as CD14 on monocytes and costimulatory markers on DCs such as CD80 and CD86 for maturation [124]. As DCs are the professional antigen-presenting cells (APCs) that are critical for antitumor activity, this impairment of DCs by TAM limits the effectiveness of cancer therapy. However, it is also known that DCs in cancer are largely defective in their function [124]. TAM can also affect certain types of DCs differently, such as, for example, follicular DCs (FDCs).

In breast cancer, FDCs expressing ER were located predominantly in germinal centers surrounding malignant foci [125]. Treatment with TAM, however, further enlarged the germinal centers, with an abundance of ER-positive FDCs, indicating the stimulatory effect of TAM on FDCs and the occurrence of TAM resistance [126]. From another perspective, the stimulatory effect of TAM on DCs can be observed on CLEC10A, a member of the C-type lectin family, which is expressed on DCs and macrophages and acts as a receptor for damaged cells. Hormone depletion by TAM induced CLEC10A ligands on damaged cells, and, in breast cancer patients, positivity of CLEC10A in tumor tissue is associated with improved disease-free and overall survival [127].

## 6. Future Directions for TAM in Investigating Tolerogenic Cells

Since TAM is broadly used in breast cancer treatment, our review has been focused mostly on the effects of TAM on the tolerogenic cells within the TME. TAM, which is known as SERM, has become one of the standard tools in treating patients with positive ER to help lower the risk of developing breast cancer. It can be used to treat women with breast cancer who have or have not gone through menopause [128]. However, being a selective modulator, TAM has its own limitations in providing potent cures within breast cancer. According to reports, TAM therapy has a negative agonistic effect on menopausal breast cancer patients by raising their risk of uterine cancer and thromboembolism [129]. Hence, this has led to TAM being revoked as a gold standard in breast cancer treatment. Aromatase inhibitors (AIs) have now surpassed TAM as the first-line therapy in breast cancer treatment [130].

In regard to tolerogenic cells, TAM administration has been observed to directly affect the MDSCs and DCs. This is because TAM has been given directly to the patient as breast cancer treatment in correlation to their responses within the immune cells. Nevertheless, in Tregs, and specifically on Foxp3, TAM has been used indirectly through the use of the CreER model in order to target specific immune checkpoints. Despite its secondary role, TAM administration is vital to the Foxp3-deficient model in order for certain genes to be targeted by the CreER model. However, even though the involvement of TAM is vital to the CreER model, there are no/very few studies focusing on the rising immune checkpoints, which are LAG-3, TIM-3, TIGIT and TNFR2. Only CTLA-4 and PD-1 are well-established immune checkpoints using the CreER model. In consequence, there is a lack of information on the effects of TAM within the mentioned immune checkpoints, mainly within the TME. Therefore, in order to discover potential therapies in cancer, there should be more studies focusing on these immune checkpoints using the CreER model.

However, is it necessary to keep researching the effect of TAM if it is no longer considered the gold standard? The answer is yes. The gold standard can never be enough within medical research, especially in cancer therapy. Cancer is a disease driven by DNA mutations, in which, within time, the cancer cells themselves can evolve and escape the current cancer therapy [64]. Hence, there should be more research conducted to understand further the effects of TAM within the rising immune checkpoints, which could potentially become cures for cancer without any resistance.

## 7. Conclusions

Here, we have discussed the effects of TAM specifically on immune tolerogenic cells. Overall, the effects can be divided into two tables. First, TAM shows it effects indirectly via Foxp3 Treg cells (Table 3) and directly through MDSCs and DCs (Table 4).

Of note, there are many limitations of the present study that need to be addressed. Clinical investigations in the present study were retrospective and not widely explored. Furthermore, due to the rarity of research on the effects of TAM within rising immune checkpoints, MDSCs and DCs, we were unable to investigate the impact of immune tolerance within the TME broadly. Thus, a collaborative multi-institutional investigation, particularly in a prospective setting, is desirable. These observations outline a crucial aspect in cancer as the effect of TAM also points out potential immune checkpoints for cancer therapy. Since new drug discovery is a long and costly process, the need to fully understand the mechanisms and effects of existing drugs such as TAM is significant and can be achieved through new investigations and innovative modifications.

## Figures and Tables

**Figure 1 biology-11-01225-f001:**
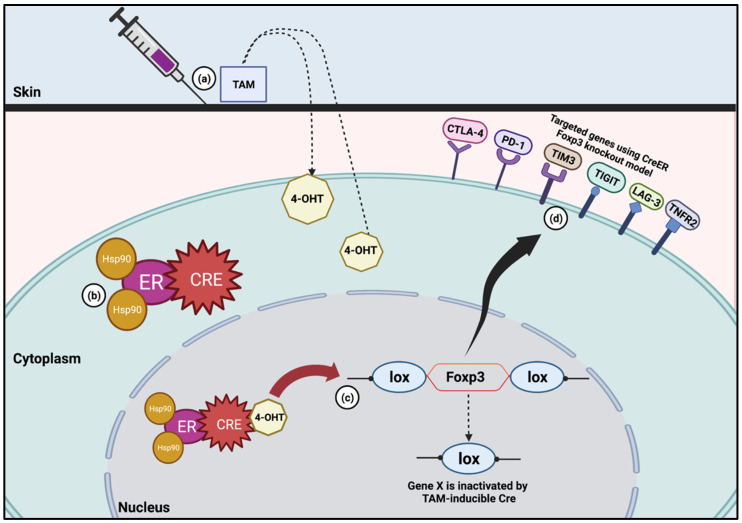
TAM-inducible CreER system in Foxp3 knockout model. (**a**) Once TAM is administered, it is metabolized into 4-hydroxytamoxifen (4-OHT). (**b**) Without TAM, CreER will result in the shuttling of the mutated recombinase into the cytoplasm. The protein will stay in this location in its inactivated state until TAM is given. (**c**) 4-OHT then binds to the ER and results in the translocation of the CreER into the nucleus, where it is then able to cleave the lox sites. (**d**) Foxp3 knockout model has been used to study specific immune checkpoints such as CTLA-4, PD-1, TIM3, TIGIT, LAG-3 and TNFR2.

**Figure 2 biology-11-01225-f002:**
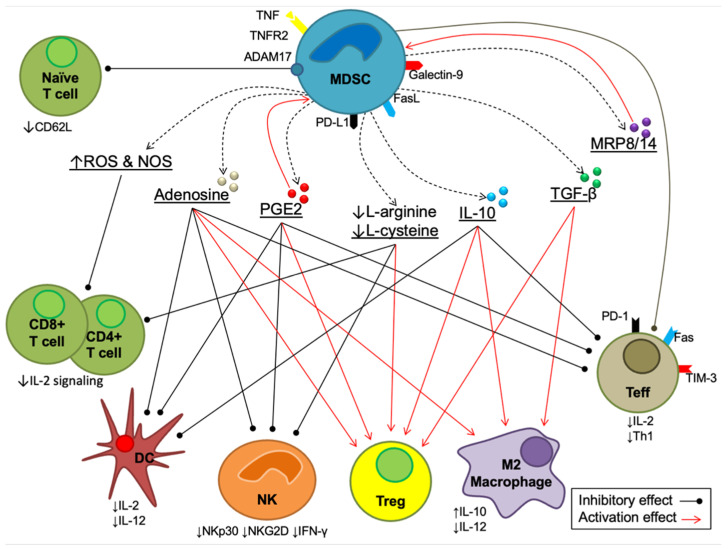
The immunosuppressive mechanisms mediated by MDSCs. There are four main mechanisms, as mentioned above. Each mechanism is explained in full in the following sections.

**Table 1 biology-11-01225-t001:** Four molecular subtypes of breast cancer.

Molecular Subtypes	Immunohistochemical Characterization
Luminal A	ER+ and (or) PR+, HER-2− and Ki-67 < 14%
Luminal B	ER+ and (or) PR+, HER-2− and Ki-67 ≥ 14% ER+ and (or) PR+, HER-2+, any level of Ki-67
HER-2 Overexpression	ER−, PR−, HER+, any level of Ki-67
Triple-Negative Type	ER−, PR−, HER−, and Ki-67 any level

ER = Estrogen Receptor. PR = Progesterone Receptor. HER2 = Human Epidermal Growth Factor Receptor 2. Ki-67 = Ki-67 Protein. ER, PR and HER2 are characterized as either positive (+) or negative (−), while Ki-67 is referred to by its percentage score.

**Table 2 biology-11-01225-t002:** Central and peripheral immunologic tolerance.

	Central Tolerance	Peripheral Tolerance
Features	Inactivation of cells required for initiation of an immune response.	Inhibition of expression to the immune response.
Site of tolerance induction	Generative lymphoid organs.	Peripheral lymphoid tissues.
Site of involvement	Afferent limb of the immune response, which is concerned with sensitization and cell proliferation.	Afferent limb of immune response, which is concerned with the generation of effector cells.
B cell participation	Immature B cells.	Mature B cells.
T cell participation	Immature thymocytes.	Mature T cells.
Mechanisms of tolerance	Clonal deletion (apoptotic cell death, negative selection).	Clonal deletion (apoptotic cell death); clonal anergy (functional inactivation without cell death); clonal ignorance (failure to recognize or recognition of antigens without costimulation); suppression of lymphocyte activation and effector functions by regulatory lymphocytes.
Function	Eliminates potentially self-reactive lymphocytes.	Maintains unresponsiveness to self-antigens.

**Table 3 biology-11-01225-t003:** Indirect effects of TAM via Foxp3 knockout model.

Immune System Involve	Specific Cells and Other Immune Checkpoints		References
Adaptive	Foxp3+ Treg	- Developing Foxp3 knockout model. - Allowing nuclear translocation of the fusion protein, excision of the floxed STOP cassette and constitutive and heritable expression of YFP in a cohort of cells that express Foxp3.	[43,54]
	CTLA4	- Act as a biological rheostat on conventional dendritic cells (cDC) by controlling the engagement with costimulatory proteins CD80 and CD86 via CTLA-4 or CD28.	[30]
	PD-1	- Immune suppression of Tregs via the reduced signaling of PI3K–AKT pathway as a mechanism.	[68]
	LAG-3	- Allows T cells to regain their cytotoxic function and potentially inhibit tumor growth.	[77]
	TIM3	- Enhances antitumor immunity, with increased expression of interleukin-10 (IL-10) and a shift to a more glycolytic metabolic phenotype.	[53]

**Table 4 biology-11-01225-t004:** Direct effects of TAM via MDSCs and DCs.

Immune System Involve	Specific Cells and Other Immune Checkpoints		References
Innate	MDSCs	- Activate several pro-inflammatory pathways in human neutrophils, including chemotaxis, phagocytosis and neutrophil extracellular trap (NET) formation. - Potent inducers of NETosis. - Enrich NK and natural killer T (NKT) cells. - Increase levels of Mo-MDSCs.	[12,115]
	DCs	- Impair differentiation and activation of DCs, limiting the effectiveness of cancer therapy. - TAM resistance due to enlargement of the germinal centers with an abundance of ER-positive FDCs.	[18,103,124,125,127,131]

## Data Availability

Not applicable.

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
