# Peer review of "The Effects of Tamoxifen on Tolerogenic Cells in Cancer"

_biology, 2022, doi:10.3390/biology11081225_

Round 1

Reviewer 1 Report

Tamoxifen (TAM) is a selective estrogen receptor modulator (SERM). It has been used to treat different stages of hormone receptor-positive breast cancer for more than two decades. TAM plays its action in two ways; it blocks the effects of estrogen in breast tissue by competing with estradiol (E2), and it binds to DNA to inhibit carcinogenesis. Furthermore, it has been suggested that TAM plays a role as immunoprotective in addition to its role in cancer therapy. However, a detailed mechanism of how TAM plays a role in modulating the tolerogenic cells is unclear. In this review, the authors have shown that TAM exhibits its direct effect on MDSC and DC and indirect effect on Foxp3 Tregs cells. The draft is well written and provides substantial graphical details to make reading and understanding easy. Unfortunately, the authors could not emphasize clinical outcomes as this area remained poorly explored. However, it is expected that research efforts and innovation in the future will present a broader picture of the many potential roles of TAM.

The presented topic matches the journal's scope. Since the draft is very well written, I do not have any significant comments to be addressed. However, I found a few grammatical /typological errors that must be corrected before formal acceptance.

Minor comments

1.     Line 35: "TME-Tregs" should be expanded at least here and later; the abbreviation can be used.

2.     The labeling in figures, Figure 1 and Figure 2 does not look consistent. It would be better to use identical fonts in both the figure         

3.     Subtitle 4.1: The font in this subtitle's text is inconsistent with the rest of the draft.

4.     Line 568, the subtitle numbering "3" is incorrect. It should be "4."

Reviewer 2 Report

The vital effects of tamoxifen (TAM) on Tregs for precise mechanistic understanding of cancer immunotherapies were comprehensively reviewed by the authors. In total, this manuscript was well-organized. Some points should be noticed as below.

1) As to the title A Review: The effects of Tamoxifen on Tolerogenic Cells in Cancer”, what does tolerogenic cells mean?; Cancer here it was equal to Breast cancer or also includes other human tumors?; “A Review” should be deleted.

2) As to “Table 3. Indirect Effects of TAM via Foxp3 knockout model”, unfortunately, for example, it was found that the contents in Ref 75 “LAG3 limits regulatory T cell proliferation and function in autoimmune diabetes.Sci Immunol. 2017 Mar 31;2(9):eaah4569” had no correlation (words) with TAM and cancer. Therefore, the authors need to check this current paper thoroughly, and correct any irrelevant references cited in Tables or other places.

3) Line 568 “3. Myeloid-derived suppressor cells (MDSCs)” should be 4. Myeloid-derived suppressor cells (MDSCs)”
